# Freshwater Water-Quality Criteria for Chloride and Guidance for the Revision of the Water-Quality Standard in China

**DOI:** 10.3390/ijerph20042875

**Published:** 2023-02-07

**Authors:** Yajun Hong, Ziwei Zhu, Wei Liao, Zhenfei Yan, Chenglian Feng, Dayong Xu

**Affiliations:** 1School of Chemical and Environmental Engineering, Anhui Polytechnic University, Wuhu 241000, China; 2State Key Laboratory of Environmental Criteria and Risk Assessment, Chinese Research Academy of Environmental Sciences, Beijing 100012, China; 3Wetland Research Center, Jiangxi Academy of Forestry, Nanchang 330032, China

**Keywords:** chloride, water-quality criteria, water-quality standard, species sensitivity distribution

## Abstract

The chloride in water frequently exceeds the standard; directly quoting foreign water-quality criteria (WQC) or standards will inevitably reduce the scientific value of the water-quality standard (WQS) in China. Additionally, this may lead to the under- or overprotection of water bodies. This study summarized the sources, distribution, pollution status, and hazards of chloride in China’s water bodies. Additionally, we compared and analyzed the basis for setting WQS limits for chloride in China; we systematically analyzed the basis for setting the WQC for chloride in foreign countries, especially the United States. Finally, we collected and screened data on the toxicity of chloride to aquatic organisms; we also used the species sensitivity distribution (SSD) method to derive the WQC value for chloride, which is 187.5 mg·L^−1^. We put forward a recommended value for freshwater WQS for chloride in China: less than 200 mg·L^−1^. The study of a freshwater WQC for chloride is not only a key point of environmental research, but also an urgent demand to ensure water ecological protection in China. The results of this study are of great significance for the environmental management of chloride, protection of aquatic organisms, and risk assessment, especially for the revision of WQSs.

## 1. Introduction

Water is the most important part of the Earth’s ecosystem and the most precious resource in nature. The quality of water environments is closely related to the ecosystem. The chloride ion (Cl^−^) is a negatively charged chlorine atom (Cl) (CAS No. 7782-50-5, atomic mass 35.45 g/mol) that forms when the chlorine atom picks up one electron. The chlorine atom is a halogen (boiling point of 33.9 °C) and never exists in free form in the environment. Chloride ions are the most common ion in water environments. The sources of chloride ions in water are as follows: chlorine (ClO_2_), which sterilizes; water flows through chloride-containing strata; water sources are polluted by domestic sewage or industrial wastewater; in coastal areas, a large amount of seawater enters the water sources due to backwater caused by high tides. High concentrations of chloride ions in drinking water produce an unpleasant taste and harm human health. Human contact with water with high concentrations of chloride ions causes damage to the skin. Chloride ions easily to be polarized in water, and polarized chloride ions can significantly speed up the corrosion reaction. Chloride ions in water have a corrosive effect on reinforced concrete, such as that used for bridges, and accelerate the aging of buildings. Chloride ions in boiler steam have a corrosive effect on boiler pipes and turbine blades [1,2,3]. When the chloride ion content is too high in soil, the growth of plants is affected, and the sustainability of the ecological environment is destroyed. Chloride ions are indispensable in living organisms. Chloride ions play an important role in maintaining the normal function of cells and in cell proliferation, excitability regulation, immune response, and other cellular activities. At the same time, many physiological and pathological processes in biological bodies are directly related to chloride ions [4,5].

Water-quality criteria (WQC) refer to the highest acceptable concentration or level of pollutants or harmful factors in the water environment, above which the pollutant will have adverse effects on human health, aquatic ecosystems, and their useful functions [6,7,8,9,10]. The objective of WQC is to protect commercially and recreationally important aquatic organisms and other important species, such as fish, invertebrates, and plankton, from the adverse effects caused by short-term exposure to high concentrations or long-term exposure to low concentrations of pollutants [11,12,13]. Environmental protection standards provide an important basis and criteria for environmental law enforcement and management, and the scientificity and rationality of their formulation process directly affect the effect of environmental management and environmental law enforcement [14]. At present, China’s GB3838−2002 “Surface Water Environmental Quality Standard” provides provisions on chloride, but this standard mainly refers to the WQC or WQS of developed countries and has no specific protection objectives, which is not suitable for China’s regional ecological environment and current environmental management needs [8,9]. The characteristics of water pollution in China are different from those in foreign countries, and the organisms in the country are also different from those in other countries. Therefore, directly quoting foreign WQC or WQSs will inevitably reduce the scientific value of the WQS in China, which may lead to the underprotection or overprotection of water bodies [15,16]. At the same time, in recent years, the chloride in the water has frequently exceeded the standard. In order to protect the health of human bodies and of the ecosystem, it is urgent to establish a standard of chloride in water. Studying chloride freshwater WQC is not only the focus of environmental criteria research, but also an important demand for water ecological protection in China. 

At present, research on the WQC or WQSs of chloride has not been reported. This study adopted the international common species sensitivity distribution (SSD) method as the derivation method of WQC, selected native Chinese species and internationally common species as the main protection object, took chloride as the research object, studied the aquatic biological WQC value applicable to China’s freshwater environment, and compared the research results with China’s existing standard. The results of this study were compared with the existing standard in China in order to provide a theoretical basis for the revision of China’s chloride WQSs. 

## 2. Sources, Pollution Status, and Hazards of Chloride in Water Bodies

### 2.1. Sources of Chloride

Chloride is present in natural water in the form of sodium, calcium, and magnesium salts. Cl^−^ is widely distributed in natural water and is present in almost all surface waters, but the content varies widely from 10 to 20 mg/L in river water to 19,000 mg/L in seawater. The source of Cl^−^ in water bodies can be divided into naturally occurring sources and anthropogenic sources.

The two main natural sources are as follows: First, the water flows through the soil layer containing chloride, which leads to the dissolution of salt deposits and other chloride-containing sediments in water; second, the river or river water from the sea is affected by the tide, which leads to an increase in the chloride content in water. Research shows that when a water source enters 1% seawater, the chloride content increases to 190 mg/L. Anthropogenic sources mainly come from industrial wastewater discharged from the chemical, petrochemical, chemical pharmaceutical, paper, cement, soap, textile, paint, pigment, food, machinery manufacturing, and leather-tanning industries. The chloride content in industrial wastewater discharged from certain industries is shown in Table 1. This type of chloride contained in wastewater discharged from human production activities is the main source of chloride pollution in surface water. In areas of human activity, industrial wastewater and domestic sewage are important sources of chloride in water bodies. The chloride content in general urban river water is much higher than that in distant suburban river water. 

In addition, a certain amount of chloride is also contained in domestic wastewater (urine contains about 1% sodium chloride). Although the amount of chloride in domestic wastewater is low, it is also an important source of chloride pollution in surface water.

### 2.2. Concentration Distribution of Chloride

The results of previous studies have shown that the solubility of sodium chloride in water bodies is 35.9 g. The content of sodium chloride in saturated aqueous solutions of sodium chloride under standard atmospheric pressure at 20 °C is approximately 6 mol/L or 351 g/L. The International Lake and Marsh Institute considers a salinity level above 500 mg/L (0.5‰) as semi-saline water. Generally speaking, the upper limit of chloride (as Cl^−^) concentrations in freshwater is considered to be 200 mg/L (0.2‰). The concentration of chloride (as Cl^−^) in seawater is generally 19,000 mg/L (19‰), and the concentration of chloride (as Cl^−^) in seawater is approximately 100 times that of the chloride content of freshwater. The average salinity in seawater is approximately 35‰, of which sodium chloride accounts for 70% and magnesium chloride accounts for 14%. The content levels of chloride in the water bodies of some of the watersheds in China are detailed in Table 2.

### 2.3. Hazards of Chlorides

When industrial wastewater and domestic sewage containing high content levels of chloride are deposited directly into rivers, damage will occur to the natural ecological balance of the water bodies, as well as the deterioration of water-quality levels. This will potentially result in the destruction of fishery production, aquaculture and freshwater resources, and the pollution of groundwater and drinking water sources. Moreover, high-chloride concentration levels in water will cause the corrosion and soil salinization of agricultural irrigation water distribution systems, as well as hinder the growth of plants and negatively affect ecosystems. Chloride ions in water bodies also have negative effects on bridge buildings and other corrosive-prone materials, resulting in the accelerated aging of important related engineering structures and potentially dangerous situations. For example, high concentrations of chloride ions will cause adverse effects on boilers, as well as metal equipment tissue intergranular cracking and corrosion.

## 3. Domestic Chloride Water-Quality Standard Limit Value, Development Basis, and Comparative Analysis

Water chloride content is generally expressed in terms of chloride ion concentrations, which are important evaluation indicators for distinguishing between different functional water bodies, since different industries and sectors have significant differences in their WQS restrictions within China. At the present time, China’s current drinking water standard in the water quality of conventional indicators requires that the concentration levels of chloride must be less than 250 mg/L. With regards to thermal power generators and steam power equipment, the water and steam quality standards of boiler ladle furnace equipment for furnace water chloride ion concentrations are clearly defined. For example, when the boiler ladle pressure ranges between 12.7 and 15.6 Mpa, the furnace water solid alkaline agent treatment of chloride ion concentrations must be less than 1.5. However, when the pressure is higher than 15.6 Mpa, the concentration levels of chloride ions in the solid alkaline agent of furnace water must be lower than 0.4 mg/L, and the concentration levels of chloride ions in the full volatile treatments of furnace water must be lower than 0.03 mg/L. A detailed understanding of the chloride content limits in different functional water systems is of major significance to environmental management and assessment processes. In this study, samples of the domestic WQS for chloride were collected for different industries and different management departments. Then, based on the comparison results, the basic requirements for the development of WQS limits, as well as other aspects, were analyzed.

This study examined the guidelines for chloride in China’s WQS system and its environmental management, along with the overall positioning of sensory traits and general chemical indicators. The chloride concentration level guidelines for China’s current water-related standards mainly include the following: GB 3838-2002 Surface Water Environmental Quality Standards; GB 5749-2006 Drinking Water Sanitation Standards; GB/T 14848-2017 Groundwater Quality Standards; GB 5084-2021 Agricultural Irrigation Water-quality Standards; CJ 94-2005 Drinking Water Purification Water-quality Standards; CJ/T206-2005 Urban Water Supply Water-quality Standards, and so on, which restrict chloride levels within acceptable safety ranges, as detailed in Table 3. The departments which issue water-related standards mainly include the Ministry of Health, Ministry of Ecology and Environment, General Administration of Quality Supervision and Inspection, and the Ministry of Construction.

## 4. Global Water-Quality Criteria and Standards for Chloride Levels

The United States was one of the first countries to study WQC and WQS. As early as 1937, the American scholar Ellis described and recorded the lethal concentrations of 114 chemicals [17]. China has carried out research for water-quality standards (WQS) for pollutants. However, China’s water-quality criteria (WQC) research started relatively late. The initial WQC and WQS research mainly involved the collection and collation of foreign information. It was considered that a detailed understanding of the progress in the research of foreign WQC and WQS for chloride was of major significance to the formulation and revision of WQC and WQS for relevant pollutants in China. It was also believed that when combined with the actual situations in China, the development of a WQC suitable for China’s national conditions could be better achieved.

### 4.1. Water-Quality Criteria and Water-Quality Standard for Chloride Levels in the United States

#### 4.1.1. Chloride Water-Quality Criteria Studies in the United States

In 1980, the United States Environmental Protection Agency (USEPA) initially identified a methodological and theoretical system for the protection of aquatic life (WQC) [18]. In 1983 and 1985, respectively, the WQC was revised. Then, in 1985, the “*Criteria guidelines for the protection of aquatic organisms and the use of the function of the derivation of quantitative national water-quality criteria guidelines*” was released. The guidance proposed in the toxicity percentage ranking method presented the USEPA-recommended derivation of the standard method for the protection of aquatic organisms or the WQC [19]. The criteria values which had been developed using the percent toxicity ranking method included the criteria maximum concentration (CMC) and the criteria continuous concentration (CCC) of two values. The CMC considered the acute toxicity effects of aquatic organisms and the CCC considered the chronic toxicity effects of aquatic organisms.

The method first categorized each test organism into a species, and then calculated the species’ mean acute value and the species’ mean chronic value. The species were further classified, and the species’ mean acute value and species mean chronic value were calculated. Subsequently, the species were further categorized into genera, and the genus mean acute value and genus mean chronic value were calculated. Finally, the criteria value by genus was calculated, which not only considered the toxicity value of individual species but also considered the linkages between species.

The acute toxicity (LC_50_, EC_50_) data for aquatic animals in the study of WQC for chloride in the United States can be described as follows:The chloride of K^+^, Ca^2+^, and Mg^2+^ were generally considered to be more acutely toxic to aquatic animals than NaCl, and chloride in aqueous environments was generally considered to be primarily associated with Na;Only the NaCl had sufficient data to be used in the derivation of WQC;No significant relationships were found between the acute toxicity of chloride to freshwater animals and the hardness, alkalinity, or pH levels;The exposure times of 24 h and 48 h were mainly chosen, with very little change observed in the acute values from 24 h to 48 h and 96 h.

The WQC of USEPA for chloride was officially published in 1988 and presented the acute toxicity values of the selected chloride to aquatic organisms. Of the thirteen aquatic species (twelve genera) for which the acute values for chloride were available, the most toxic of the thirteen species was *Daphnia pulex*, with a toxicity value of 1470 mg/L. The least toxic was *Anguilla rostrato*, with a toxicity value of 11,940 mg/L. The most sensitive of the twelve genera was *Daphnia pulex*, with an acute toxicity of 2540 mg/L, and the least sensitive was *Anguilla rostrato*. The average acute value of the most sensitive genus was only six times higher than the average acute value of the least sensitive, with invertebrates generally being more sensitive than vertebrates. The final acute values of chloride were calculated using a percentage toxicity ranking method, and the mean acute values of the four genera with cumulative probabilities close to 0.05 were selected. The final acute value of chloride was calculated to be 1720 mg/L, taking an effect factor of 2 and a CMC of 860.0 mg/L.

The chronic toxicity of chloride to aquatic organisms was investigated by selecting typical aquatic organisms as the targeted samples. The study yielded a chronic value of 372.1 mg/L for *Daphnia magna*, with a calculated acute-to-chronic ratio (ACR) of 3.951, and a chronic value of 922.7 mg/L for *Oncorhynchus mykiss*, with an ACR of 7.31. The chronic value of chloride for *Pimephales promelas* was determined to be 433.1 mg/L, with an ACR of 15.17. The ACRs for chloride for the three species mentioned above were 7.31, 15.17, and 3.95, respectively, and the final ACR was determined using the geometric mean of the ACRs for the three species of 7.594, which was the final ACR. The final acute value divided by the final ACR resulted in a final CCC value of 226.5 mg/L (rounded to 230 mg/L).

The final WQC of the USEPA for chloride was expressed as an average four-day average concentration of dissolved chloride not exceeding 230 mg/L, and a one-hour average concentration not exceeding 860 mg/L every three years when combined with sodium. However, it should be noted that the criteria value may not provide adequate protection when chloride is combined with potassium, calcium, or magnesium. In addition, due to the narrow range of acute sensitivity to chloride in freshwater animals, a range beyond the aforementioned criteria may affect many species.

#### 4.1.2. Chloride Water-Quality Standards Studies in the United States

Since natural differences in water ecosystems cannot be identified with a uniform value for delineation, making it impossible to accurately develop a national standard which will meet the separate requirements of the nation’s waters, the United States does not have a national WQS for federally owned waters. The United States Federal Water Pollution Control Act Amendments of 1972 requires the EPA to publish WQC guidelines which accurately reflect the latest scientific research. However, the WQC is not used directly in the regulations, but only to provide a scientific basis for the development of WQS in separate states. The EPA develops guidelines to assist each state in revising the criteria presented in the document, and to develop WQS and other water-related programs of the Agency. For drinking water WQS, the USEPA recommended maximum value is 250 mg/L. The United States’ regulatory status is final. The United States’ secondary drinking water regulations include non-mandatory federal guidelines on drinking water which may have impacts on appearance (such as teeth or skin staining) or are sensory (such as taste, smell, or color).

### 4.2. Development of the Water-Quality Criteria and Standards for Chloride in Canada

The short-term threshold for chloride for aquatic organisms in Canada, which was derived from lethality studies, is 640 mg/L. The long-term threshold for chloride for aquatic organisms, assessed as no effect concentrations (NOEC) or (LOEC) for aquatic organisms using calcium chloride and sodium chloride, is 120 mg/L [20].

The chloride levels in Canadian natural surface waters are all below 10 mg/L and often below 1 mg/L. A WQS for chloride in drinking water has been established at 250 mg/L. When concentrations exceed the WQS, chloride can impart an undesirable taste to water and beverages made from water, and can lead to the corrosion of distribution systems. Therefore, the surface water limit for Canadian drinking water sources has been set at 250 mg/L [20].

## 5. The Derivation of the Water-Quality Criteria for Chloride in China

### 5.1. Toxicity Data Collection and Selection

The toxicity data of chloride used in this study mainly included widely referenced online toxicity databases, such as CNKI (http://www.cnki.com, accessed on 1 December 2022); US EPA ECOTOX DATA (https://cfpub.epa.gov/ecotox/, accessed on 1 December 2022); Web of Science (https://www.webofscience.com/wos/, accessed on 1 December 2022); and government documents. The data were selected based on their reliability, relevance, adequacy [21,22,23], and other principles. In the cases of different endpoints for the same species, the most sensitive endpoints were selected. In addition, if multiple toxicity values were available for the same endpoint and species, their geometric mean was calculated [12,13].

### 5.2. Derivation of the Water-Quality Criteria Value for Chloride

In this study, a theoretical system of WQC suitable for China’s national conditions was derived by analyzing the methodology of the WQC derivation in the United States, combined with WHO and other countries’ WQC research methods. A series of WQC research studies in China was systematically carried out, and the main theory and method of WQC research in China was finally established [8,9,14]. The “*Technical guidelines for deriving water-quality criteria for freshwater organisms*” clearly expressed the requirements of organism toxicity data for calculating WQC risk thresholds, establishing a screening method for the reliability of organism toxicity data, and bounded the coverage of the organism toxicity data. It also proposed that the focus should be on the toxicity data of China’s native species in order to accurately calculate the relevant WQC thresholds and aquatic ecological risk thresholds for pollutants [24]. It was found that when comparing other nations’ WQC derivation methods, one of the most widely recognized was the SSD method [7,22], which uses a fitted SSD curve to express the highest concentration levels which do not cause adverse effects on a particular biological group. The 5% species hazard concentrations (HC_5_) are generally used to express the concentration limits which protect at least 95% of the species. The HC_5_ value is divided by the assessment factor (AF; a range expressed from 2 to 10) [25] for the purpose of calculating the predicted no-effect value (PNEC) for chloride. In this study, a total of twenty toxicity data of chloride to aquatic organisms were collected and collated (Table 4; fourteen toxicity data in the table and another six toxicity data from the USEPA database (https://cfpub.epa.gov/ecotox/, accessed on 1 December 2022). The twenty toxicity data were tested for normal distributions, and the toxicity concentration data were normally distributed after taking logarithms, resulting in the chloride SSD curve shown in Figure 1.

The calculated chloride concentration (HC_5_ value) limit which enabled the protection of at least 95% of the species was determined to be 1875 mg/L. The short-term WQC for aquatic organisms was obtained as 937.5 mg/L when the AF was taken as 2. The long-term WQC for aquatic organisms was obtained as 187.5 mg/L when the AF was taken as 10. The results of the long-term WQC study for aquatic organisms were smaller than those of the USEPA. However, the differences in the studies included differences in the species distributions in differing geographic areas and differences in species sensitivity to different biological groups [35]. Furthermore, differences in SSD curve fitting models and dissimilar amounts of data used may have potentially affected the derived values of the WQC studies. Finally, differences in AF and ACR may have also caused changes in the WQC values [11,12].

The WQC or WQS values of chloride WQS which were derived from the above-mentioned different methodological studies are detailed in Table 5.

The chloride content of natural water varies widely. Therefore, considering the varying treatments for drinking water supply quality, with chloride potentially affecting the taste of drinking water, it has been found that people generally have a perceived concentration threshold for chloride of 210 to 230 mg/L [36,37]. According to the criteria and reference information of foreign standards, and through a comprehensive comparative analysis process, the recommended value of freshwater WQS for chloride in China is less than 200 mg/L.

## 6. Conclusions

China’s current WQC and WQS are mainly based on the development guidelines of foreign WQC. In the revisions of WQC or WQS, regional and scientific WQC guidelines should be more prominent in order to develop China’s WQS for different protection objectives and provide reasonable protection of China’s ecosystems. The current surface water environmental quality standards have made significant contributions to human health, as well as water ecological safety. However, with the rapid development of China’s society and the more in-depth study of WQC, the current standards have gradually been revealed to have some short-comings, such as difficulties in coordinating the water-quality relationships between different water systems while taking into account the multiple functions for each type of water.

In this study, toxicity data of chloride to aquatic organisms were collected and screened and a WQC value of 187.5 mg/L for chloride was derived using a species sensitivity distribution method. Subsequently, based on the analysis results, the recommended value of freshwater WQS for chloride in China was proposed to be less than 200 mg/L. In addition, through the study of WQC of chloride, this study found that the surface water environmental quality standard for chloride in China has the problem of unreasonable protection for freshwater aquatic organisms. Therefore, due to the aforementioned problem of unreasonable protection, there is an urgent need to revise the surface water environmental quality standards in China.

## Figures and Tables

**Figure 1 ijerph-20-02875-f001:**
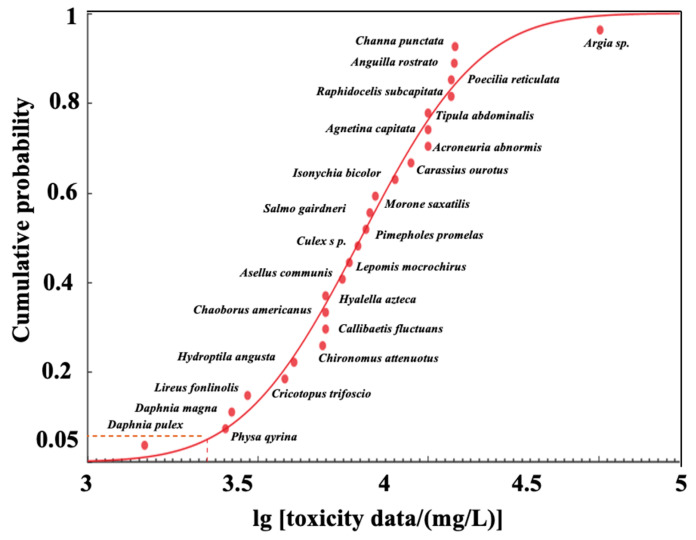
The species sensitivity distribution for chloride based on the toxicity data.

**Table 1 ijerph-20-02875-t001:** The range of chloride content in wastewater discharged by certain industries.

Industries	Sources of Wastewater	Chloride Content (Cl^−^, mg/L)	Median (Cl^−^, mg/L)
Metallurgical factory	Iron smelters wash sewage	100.0–600.0	350.0
Metallurgical factory	Nylon production sewage	475.0–3340.0	1907.5
Petrochemical industry	Synthetic rubber sewage	2670.0–2800.0	2735.0
Petrochemical industry	Butadiene sewage	1277.0–1350.0	1313.5
Petrochemical industry	Ethylene propylene rubber sewage	361.0–602.0	481.5
Printing and dyeing mill	Steam sewage	103.3–168.1	135.7
Printing and dyeing mill	Rinse the sewage	234.4–296.0	265.2
Tannery	Wastewater for ash removal	1700.0	-
Tannery	Chromite tanning wastewater	215,000.0	-

**Table 2 ijerph-20-02875-t002:** Chloride content levels in the water bodies of some watershed areas in China.

Area	Type of Water	Chloride Content (Cl^−^, mg/L)	Median (Cl^−^, mg/L)
Luoyang City	Tap water	24.01–42.97	33.49
	Recycling water	22.37–47.55	34.96
	Underground water	52.97–59.49	56.23
	Surface water	12.07–52.97	32.52
Yangtze estuary water	Surface water	45.16–178.11	111.42
Qiantang Estuary	Surface water	12.70–48.50	30.60
Minjiang River Estuary	Water plant intake	132.00–977.00	435.50
Nandu River	Surface water	17.44–9564.80	487.12
Kanazawa Reservoir	Surface water	42.00–52.42	47.21
Hun River	Surface water	76.77–94.21	85.49
Liao River	Surface water	47.09–64.77	55.93

**Table 3 ijerph-20-02875-t003:** Comparison of the water-quality limits and emission limits for chloride in China’s current standards.

Standards	Category	Limit mg/L
GB 3838-2002	Standard limit of supplementary projects of centralized domestic water source of surface water	250
GB 5749-2006	Water-quality routine indexes and limits/sensory traits and general chemical indexes	250
	Partial water-quality indicators and limits/sensory traits and general chemical indicators for small centralized and decentralized water supplies	300
GB/T 14848-2017	Groundwater quality classification index class I	≤50
	Groundwater quality classification index class II	≤150
	Groundwater quality classification index class III	≤250
	Groundwater quality classification index class IV	≤350
	Groundwater quality classification index class V	>350
GB 5084-2021	Basic control project standard value of irrigation water quality/water farming	350
	Basic control project standard value of irrigation water quality/dry farming	350
	Basic control project standard value of irrigation water quality/vegetable	350
CJ 94-2005	Drinking water-quality standards/general chemical index limits	100
CJ/T206-2005	Routine inspection items of water quality of urban water supply and limited/sensory characters and general chemical indexes	250

**Table 4 ijerph-20-02875-t004:** Toxicity data of chloride to aquatic organisms.

Genus	Species Name	Species Latin Name	Toxic Effect	Endpoint	Exposure (h)	SGMV (mg/L)	References
Algae	Green Algae	*Raphidocelis subcapitata*	Physiology	EC_50_	96	11,688.56	[26]
Crustaceans	Aquatic Sowbug	*Asellus communis*	Mortality	LC_50_	24	5600	[27]
Crustaceans	Water Flea	*Daphnia magna*	Intoxication	Immobile	48	4200	[28]
Crustaceans	Scud	*Hyalella azteca*	Mortality	LC_50_	96	5000	[29]
Fish	Snake-Head Catfish	*Channa punctata*	Mortality	LC_50_	96	12,000	[30]
Fish	Striped Bass	*Morone saxatilis*	Mortality	LC_50_	24	7000	[31]
Fish	Guppy	*Poecilia reticulata*	Mortality	LC_50_	96	11,700	[32]
Insects/Spiders	Common Stonefly	*Acroneuria abnormis*	Mortality	LC_50_	96	10,000	[33]
Insects/Spiders	Stonefly	*Agnetina capitata*	Mortality	LC_50_	96	10,000	[33]
Insects/Spiders	Damselfly	*Argia sp.*	Mortality	LC_50_	24	32,000	[27]
Insects/Spiders	Mayfly	*Callibaetis fluctuans*	Mortality	LC_50_	96	5000	[29]
Insects/Spiders	Midge	*Chaoborus americanus*	Mortality	LC_50_	96	5000	[29]
Insects/Spiders	Mayfly	*Isonychia bicolor*	Mortality	LC_50_	24~72	8000	[34]
Insects/Spiders	Crane Fly	*Tipula abdominalis*	Mortality	LC_50_	96	10,000	[33]

Note: In the table, SGMV indicates the species geometric mean value; EC_50_ represents 50% of the effective concentration; LC_50_ represents 50% of the lethal concentration.

**Table 5 ijerph-20-02875-t005:** Comparison of chloride water-quality criteria/standard values.

Countries	Research Method	Criteria/Standard (mg/L)	References
USEPA	Toxicity percentage ranking method	CMC: 860 CCC: 226.5	[19]
Canada	Sense	250	[20]
This study	SSD	Short-term WQC: 937.5 Long-term WQC: 187.5	This study

## Data Availability

The data presented in this study are available on request from the corresponding author upon reasonable request.

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
