# Peer review of "Freshwater Water-Quality Criteria for Chloride and Guidance for the Revision of the Water-Quality Standard in China"

_ijerph, 2023, doi:10.3390/ijerph20042875_

Round 1

Reviewer 1 Report

The manuscript tilted “Study on the water quality criteria for chloride and its enlightenment to the revision of water quality standard in China” reviewed and summarized the current research progress of WQC for chloride in different countries, and discussed the methodology exploration for deriving WQC of chloride. Moreover, the PNEC value of chloride using toxicity data were derived. The results of this study are of great significance for environmental management of chloride, protection of aquatic organisms and risk assessment, especially for the revision of WQS.

In general, the paper was a well-organized review article. The structures of article are sound and descriptions of the contents are clear. I recommend accepting the paper after some minor revisions.

1.        The title should be revise as “Study on the freshwater water quality criteria for chloride and its enlightenment to the revision of water quality standard in China”.

2.        Line 19-20: “The basis for setting WQC for chloride in foreign countries were analyzed systematically.” The number of characters is over the limitation.

3.        Line 28-36: The sentence should be shortened to improve readability.

4.        The term of “chloride” should be defined in Introduction.

5.        Line 183: “The water quality criteria and standard for chloride in other countries” should be revise to “The water quality criteria and standard for chloride in abroad”.

6.        Line 193: “4.1. The United States chloride water quality criteria and standard development” should be “4.1. The water quality criteria and water quality standard of chloride in United States”.

7.        Line 280: “5.1. Chloride toxicity data collection and screening” revise to “Toxicity data collection and selection”.

8.        Line 311: “is” should be “was”.

9.        Line 313: Write the full name of EC50 and LC50.

10.    Line 331: please provide references.

11.    The authors need to re-check carefully the content of the review and make sure that all the contents are adequate and helpful for other researchers. Some mistakes also need to be revised carefully.

Reviewer 2 Report

The article entitled “Study on the water quality criteria for chloride and its enlightenment to the revision of water quality standard in China” summarized the current research progress of WQC for chloride in different countries, and discussed the methodology exploration for deriving WQC of chloride. The research problem is hot and cutting-edge, which has certain scientific significance, but there are also some problems.

1. In abstract: “The characteristics of water pollution in China are obviously different from other countries, and the organism is also different from other countries. Therefore, directly quoting foreign water quality criteria (WQC) or standard will inevitably reduce the scientific of water quality standard (WQS) in China, which may lead to insufficient or over-protection of water bodies. At the same time, in recent years, the chloride in water exceeds the standard frequently. In order to protect the health of human body and ecosystem, it is urgent to establish the standard of chloride in water”, this sentence is too long, rewrite it.

2. In key words: Please give the full write of the key word “SSD”.

3. The concept of chloride is rather general, please give the exact definition.

4. “Selects Chinese native species and international common species”, what is the mean of “native species”.

5. “Water chloride content is usually expressed in terms of chloride ion concentration, chloride ion concentration as an important evaluation indicator to distinguish between different functional water bodies, in different industries in China, different sectors of its WQS have significant differences in the restrictions.”, re-write this sentence.

6. It is recommended to delete the blank rows in Table 3

7. In the section “4. The water quality criteria and standard for chloride in other countries”, why did the authors only compare the United States and Canada? Do other countries have water quality criteria for chloride?
